# Mitochondrial Metabolism in T-Cell Exhaustion

**DOI:** 10.3390/ijms26157400

**Published:** 2025-07-31

**Authors:** Fei Li, Yu Feng, Zesheng Yin, Yahong Wang

**Affiliations:** 1Institute of Pathogen Biology, School of Basic Medical Sciences, Lanzhou University, Lanzhou 730000, China; fyu2024@lzu.edu.cn (Y.F.); yinzsh2023@lzu.edu.cn (Z.Y.); 2School of Public Health, Lanzhou University, Lanzhou 730000, China; wyahong2023@lzu.edu.cn

**Keywords:** T-cell exhaustion, mitochondria, metabolism, metabolic reprogramming, mitochondrial dynamics

## Abstract

T cells play a vital role in resisting pathogen invasion and maintaining immune homeostasis. However, T cells gradually become exhausted under chronic antigenic stimulation, and this exhaustion is closely related to mitochondrial dysfunction in T cells. Mitochondria play a crucial role in the metabolic reprogramming of T cells to achieve the desired immune response. Here, we compiled the latest research on how mitochondrial metabolism determines T cell function and differentiation, with the mechanisms mainly including mitochondrial biogenesis, fission, fusion, mitophagy, and mitochondrial transfer. In addition, the alterations in mitochondrial metabolism in T-cell exhaustion were also reviewed. Furthermore, we discussed intervention strategies targeting mitochondrial metabolism to reverse T cell exhaustion in detail, including inducing PGC-1α expression, alleviating reactive oxygen species (ROS) production or hypoxia, enhancing ATP production, and utilizing mitochondrial transfer. Targeting mitochondrial metabolism in exhausted T cells may achieve the goal of reversing and preventing T cell exhaustion.

## 1. Introduction

During many chronic infections, such as lymphocytic choriomeningitis virus (LCMV), human immunodeficiency virus (HIV), hepatitis B virus (HBV), hepatitis C virus (HCV), *Mycobacterium tuberculosis* (*M. tuberculosis*), and cancer, persistent antigen exposure can lead to dysfunction or even exhaustion of antigen-specific T cells [1]. Exhausted T cells exhibit overexpression of multiple inhibitory receptors, such as PD-1 and T cell immunoglobulin mucin 3 (TIM-3), and lymphocyte activation gene 3 (LAG-3), along with a gradual reduction in effector function and proliferative capacity, and a progressive loss of memory T cell potential, including the antigen-independent self-renewal ability and the capacity to generate strong recall responses [2]. Additionally, exhausted T cells exhibit altered transcription factor expression, including upregulated expression of B lymphocyte-induced maturation protein-1 (Blimp-1) and thymocyte selection-associated HMG box (TOX) [3,4]. Persistent antigen stimulation, high levels of reactive oxygen species (ROS), and hypoxia are key drivers of T-cell exhaustion [5,6]. Immune checkpoint blockade (ICB) therapy, such as nivolumab, can rejuvenate exhausted T cells; however, it still has many limitations [7,8]. Therefore, it is necessary to explore new mechanisms of T cell exhaustion, particularly the role of mitochondrial metabolism in its regulation, to identify new therapeutic targets for treating T cell exhaustion.

The immune status of T cells is determined by their metabolic fitness, leading to different functional abilities and disease outcomes [9]. During acute infection, naïve T cells undergo metabolic reprogramming from mitochondrial oxidative phosphorylation (OXPHOS) towards aerobic glycolysis to provide a quick energy source and meet the increased bioenergetic requirements of effector T cells [10,11]. After effector T cells undergo a contraction stage, a small number of memory T cells persist, and these transformed memory T cells are characterized by their utilization of mitochondrial OXPHOS and fatty acid oxidation (FAO) for energy to sustain a recall response during reinfection [12]. During chronic infection, continuous antigenic stimulation drives T cell dysfunction and even exhaustion. Exhausted T cells can be divided into two subsets: progenitor exhausted T cells and terminally exhausted T cells [13]. Progenitor exhausted T cells are a type of “stem-like” T cell population with self-renewal capacity that responds to PD-1 pathway blockade therapy, while terminally exhausted T cells exhibit impaired proliferative ability and do not respond to PD-1 blockade [14,15,16]. Progenitor exhausted CD8^+^ T cells exhibit intermediate expression of PD-1, high expression of CD127, chemokine receptor CXCR5, and T-cell factor 1 (TCF-1), whereas terminally exhausted T cells show high expression of PD-1 and TIM-3, with a loss of expression of TCF-1 and CXCR5 [17]. CXCR5^+^ CD8^+^ T cells also express several genes related to the self-renewal and maintenance of hematopoietic stem cells in the Wnt signaling pathway [18]. Metabolically, progenitor exhausted T cells use mitochondrial FAO and OXPHOS for energy [19], whereas terminally exhausted T cells primarily rely on glycolytic metabolism, with impaired glycolysis and OXPHOS [20,21,22]. The exhausted T cells mentioned in this review mainly refer to terminally exhausted T cells.

Mitochondria are the energy factories of cells, and mitochondrial activity plays a critical role in the activation and maintenance of antigen-specific responses, particularly during memory responses and T-cell exhaustion [23]. Mitochondrial dysfunction is a hallmark of T-cell exhaustion. The mitochondrial metabolic capacity can be an important factor to consider when designing immunotherapies to rescue exhausted T cells in cancer or chronic infections [24]. Therefore, studying the changes in mitochondrial metabolism during exhaustion can help identify new targets for therapeutic intervention.

## 2. Mitochondrial Metabolism Determines T Cell Function and Fate

Mitochondria are widely recognized as the powerhouse of cells, producing adenosine triphosphate (ATP) through OXPHOS to provide energy for cellular functions. They serve as the primary metabolic regulators of T cells [24]. Mitochondrial function and energy metabolism influence T cell function, activation, proliferation, differentiation, memory, and exhaustion [25]. Furthermore, mitochondrial metabolism determines T-cell fate and function through metabolic reprogramming [24,26,27]. It is closely linked to mitochondrial morphology, which is affected by mitochondrial biogenesis, fission, fusion, mitochondrial autophagy (mitophagy), and mitochondrial transfer [28,29] (Figure 1).

### 2.1. Mitochondrial Biogenesis Facilitates T Cell Metabolic Reprogramming

Mitochondrial biogenesis is an important part of mitochondrial quality control [30]. Mitochondrial biogenesis occurs rapidly in nascent activated CD8^+^ T cells and is crucial for supporting cytokine generation by T cells in early immune responses [31]. Proliferator-activated receptor γ coactivator 1 α (PGC-1α) is a critical regulator of mitochondrial biogenesis and mitochondrial plasticity [32]. PGC-1a is located upstream of the mitochondrial biogenesis system and serves as a junction between mitochondrial external stimulus signals and internal regulation. PGC-1α activates peroxisome proliferator-activated receptor α (PPAR-α) and acts as a coactivator of PPAR-α in the transcriptional regulation of mitochondrial FAO capacity [33]. Overexpression of PGC-1α enhances mitochondrial biogenesis, restores T cell function, and improves anti-tumor immunity [21].

### 2.2. Mitochondria Dynamics Control T Cell Fate

Mitochondrial cristae morphology reflects metabolic states [34]. Mitochondria are highly dynamic organelles that maintain normal counts, shape, and function through constant fusion and division, a process commonly referred to as mitochondrial dynamics [35]. Fusion relieves stress by mixing the contents of partially damaged mitochondria, while fission helps to remove damaged mitochondria, promotes apoptosis under high cellular stress, and is necessary for generating new mitochondria [36]. Mitochondrial fusion is associated with increased OXPHOS and ATP generation [37]. Furthermore, the dynamics of mitochondrial fusion and fission are controlled by proteins that regulate mitochondrial form. For example, fission is governed by the mitochondrial fission factor (Mff) and dynamin-related protein 1 (Drp1), whereas optic atrophy 1 (OPA1), mitofusin 1 (MFN1), and mitofusin 2 (MFN2) favor mitochondrial fusion and promote cellular ATP generation [30,38,39].

Mitochondrial dynamics determine T cell fate through metabolic programming [26]. Effector T cells possess fissed mitochondria with loosened cristae, whereas memory T cells have fused mitochondria with tighter cristae and expanded space. Altering mitochondrial fission/fusion drives changes in cristae morphology, further inducing metabolic programming and ultimately controlling T cell differentiation. Memory T cells are characterized by enhanced mitochondrial function [40]. Forcing mitochondrial fusion drives memory T cell differentiation by promoting OXPHOS and FAO, while mitochondria fission facilitates aerobic glycolysis within effector T cells [26,41]. Furthermore, mitochondrial fission produces discrete and fragmented mitochondria, which contribute to elevated ROS production and promote mitochondrial polarization [42,43]. Under normal circumstances, the small amount of ROS generated by mitochondria is crucial for promoting T cell activation and proliferation to acquire effective functions. Excessive mitochondrial ROS can activate continuous nuclear factor of activated T cells (NFAT) signaling [23], leading to terminal exhaustion [5]. Since Drp1-mediated fission promotes mitophagy, decreased Drp1 activity or DRP1 deletion may lead to T cell exhaustion by inhibiting mitophagy [44]. Interestingly, there are also opposing views on the role of Drp1. Specifically, Drp1 deficiency leads to mitochondrial fusion, while Drp1 knockout drives the conversion from effector T cells to a memory phenotype [45]. For instance, the treatment of T cells with ‘mitochondrial fission inhibitor’ mdivi-1 [46] and the ‘fusion promoter’ M1 [47] induces mitochondrial fusion, thereby conferring a memory T cell phenotype and promoting the generation of memory-like T cells [26]. Therefore, inhibiting fission may prevent ROS production and favor the formation of memory T cells [43,48]. The role of Drp1 in exhaustion and memory formation remains to be explored.

### 2.3. Mitochondrial Autophagy (Mitophagy) Maintains Cellular Homeostasis

Mitophagy, the process of selectively eliminating dysfunctional mitochondria, is essential for mitochondrial quality control [49]. Damaged mitochondria are often accompanied by a reduction in electron transport chain (ETC) efficiency, which then leads to a decrease in ATP generation [50]. Dysfunctional mitochondria or elevated depolarized mitochondria are the main source of ROS that cause oxidative damage and lead to cell necrosis [51,52]. However, mitophagy removes ROS to maintain mitochondrial integrity [53]. Repressing mitophagy results in excessive ROS accumulation and causes apoptosis in cells [54]. Mitophagy is mediated by the PTEN-induced putative protein kinase 1 (PINK1)/Parkin-directed pathway [55,56]. Park2 is an E3 ligase that also mediates mitophagy, which can destroy mitophagy activity after removal and further promote the acquisition of T cell mitochondrial depolarization phenotype. Consistently, the blockade of mitophagy by oligomycin plus the mitophagy inhibitor mdivi-1 also favors the formation of depolarized mitochondria, which reinforces T-cell exhaustion [57]. In addition, mitophagy is vital for memory formation, primarily by eliminating dysfunctional mitochondria and maintaining mitochondrial homeostasis [58,59].

### 2.4. Mitochondrial Transfer Modulates Intercellular Communication

Mitochondrial transfer/transplantation is the process of transporting healthy/functional mitochondria from donor cells into surrounding cells with damaged mitochondria, thereby increasing mitochondrial mass and improving functionality [60,61,62]. Currently, there are four main modes of mitochondrial transfer, including extracellular vehicles (EVs), endocytosis, gap junctions, and tunneling nanotubes (TNTs) [29,63,64]. On one hand, cancer cells could deprive mitochondria of T cells through intercellular nanotubes-TNTs to evade the immune system and enhance cancer aggressiveness [65]. In this regard, mitochondrial transfer seems to be a detrimental phenomenon for tumors. On the other hand, T cells also acquire mitochondria from other cells, promoting cell proliferation and differentiation, suggesting that mitochondrial transfer may be a novel strategy for treating chronic infections [66]. Altogether, mitochondrial transfer functions as a double-edged sword for T cells. However, a major limitation currently hindering the translation of mitochondrial transfer into clinical practice is its low efficiency (about 10%, up to 28%) and the lack of a simple method for tracking transferred mitochondria [67].

## 3. Changes in Mitochondrial Metabolism in T-Cell Exhaustion

The metabolic characteristics of exhausted T cells include disrupted mitochondrial energy and respiration, increased depolarized mitochondria, and accumulation of ROS, leading to reduced glucose uptake, defective glycolysis, and OXPHOS [6]. PD-1 signaling promotes these metabolic alterations by driving Blimp1-mediated inhibition of PGC-1α, a major regulator of mitochondrial biogenesis. The repression of PGC-1α is particularly prominent in viraemic HIV-1-infected exhausted CD8 T cells [68]. Inhibition of PGC-1α leads to elevated mitochondrial ROS, which acts as a phosphatase inhibitor, activates phosphotyrosine signaling, and induces NFAT localization, thereby promoting the activation of exhaustion-associated genes, such as *TOX* and *Blimp1*. Enforcing the expression of PGC-1α can reverse PD-1-induced bioenergetic deficiency and rescue exhausted T cells [6].

Exhausted T cells display a reduction in abnormal mitochondrial morphology and mass, an increase in mitochondrial size, a decrease in mitochondrial membrane potential, and a loss of mitochondrial fitness [69], indicating mitochondrial dysfunction. This contributes to the inability of exhausted T cells to effectively utilize OXPHOS for energy [22]. Compared to memory T cells, exhausted T cells maintain a slightly larger mitochondrial mass and significantly increased amounts of depolarized mitochondria [70]. Mechanically, oxidative stress caused by mitochondrial dysfunction antagonizes the proteasomal degradation of hypoxia-inducing factor 1α (HIF-1α), which mediates the glycolytic reprogramming of progenitor exhausted T cells towards terminally exhausted T cells [71]. Additionally, PD-1 signaling induces alterations in cristae morphology, such as reducing the number and length of mitochondrial cristae, and decreasing the expression of *CHCHD3* and *CHCHD10*, which are involved in cristae structure and organization, ultimately leading to apparent mitochondrial dysfunction [72]. PD-1 signaling directly decreases mitochondrial fission by downregulating Drp1 phosphorylation [73].

Additionally, ATP is not only necessary for the functions of activated T cells, such as cell proliferation and cytokine production, but also for the homeostasis of mitochondria [74]. Continuous antigen stimulation disrupts T cell OXPHOS, driving exhaustion through impaired ATP generation and limited self-renewal gene expression [75]. During chronic infection, ATP is reduced and insufficient, thereby unable to maintain mitochondrial function and mass, leading to T cell dysfunction. CD39, also known as external nucleoside triphosphate diphosphate hydrolase-1 (ENTPD1), is the prototype of the ENTPDase family and catabolizes extracellular pro-inflammatory ATP and ADP into AMP. AMP is further degraded by the ecto-5′-nucleotidase CD73 into anti-inflammatory adenosine, which is subsequently converted to inosine by adenosine deaminase [76,77]. CD39 is highly expressed on exhausted T cells and is defined as a new marker of T cell exhaustion [78,79]. CD73 has also emerged as a promising therapeutic target.

Furthermore, the exhausted T cells also exhibit significant downregulation of mitochondrial genes encoding specific mitochondrial proteins [80]. These genes encode proteins, including mitochondrial fusion protein OPA1 and carnitine palmitoyl transferase 1α (CPT-1α), which is involved in controlling mitochondrial FAO and driving memory T cell formation [81]. Exhausted CD8^+^ T cells display various structural or functional mitochondrial changes, which are caused by impaired mitochondrial mitophagy capacity [57]. In addition, the mitochondrial dysfunction in exhausted T cells is associated with the depletion of mitochondrial transcriptional factor A (Tfam), which leads to dysfunction. Tfam deficiency in T cells can result in severe mitochondrial DNA loss, disrupting T cell receptor (TCR)-driven cell proliferation and effector function [82,83]. Moreover, mitochondrial phosphatase PTPMT1-mediated metabolism is necessary to maintain the differentiation and expansion of effector T cells. Loss of PTPMT1 impairs T cell anti-tumor immunity and accelerates T cell exhaustion [84].

## 4. Potential Therapeutic Strategies for Mitochondrial Metabolic Regulation in Reversing T-Cell Exhaustion

Exhausted T cells exhibit a repressed mitochondrial metabolism and decreased mitochondrial mass. Correcting mitochondrial dysfunction and restoring mitochondrial function has been shown to reinvigorate exhausted T cell function [80,85]. Restoring mitochondrial mass can improve cell function. Furthermore, enhancing mitochondrial function might represent a therapeutic strategy for reversing exhausted T cells [21]. Several strategies, such as inducing PGC-1α expression, alleviating ROS production or hypoxia, inducing ATP production, and utilizing mitochondrial transfer, may be effective (Table 1).

### 4.1. Inducing PGC-1α Expression

Enforcing mitochondrial biogenesis is beneficial for boosting T cell immune response, and promoting the biogenesis and function of mitochondria can prevent T cell exhaustion [86]. In combination with immune checkpoint inhibitors, enhancing mitochondrial biogenesis may enhance the efficacy of immunotherapy and combat immune exhaustion [80]. PGC-1α enhances mitochondrial biogenesis [87]. Bezafibrate, a mitochondrial activator and PGC-1α agonist, improves the antitumor response to PD-1 blockade nivolumab [88]. Another mitochondrial regulator, complement C1q binding protein (C1QBP), contributes to mitochondrial biogenesis via PGC-1α. Insufficient C1QBP can result in compromised mitochondrial fitness in T cells, exacerbating T cell exhaustion, while the overexpression of C1QBP may modify the state of exhaustion [89]. Overexpression of PGC-1α promotes memory formation [90] and restores mitochondrial function and exhausted CD8 T cell function by reversing metabolic dysregulation [6,21].

Furthermore, the costimulatory molecule 4-1BB promotes PGC-1α-dependent mitochondrial fusion and biogenesis by activating p38-MAPK and enhancing metabolism [91]. 4-1BB costimulation can enhance mitochondrial fusion and biogenesis [92]. Overexpression of 4-1BB increased mitochondrial mass and transmembrane potential, thereby enhancing mitochondrial respiratory capacity [93]. Moreover, 4-1BB agonists enhance T cell activity in a PGC-1α-dependent manner, induce a memory-like state, and metabolically facilitate the anti-PD-1 response [92]. Coenzyme Q10 (CoQ10) transfers electrons from complexes I and II to complex III, a crucial step in ATP production. The concentration of CoQ10 is regarded as an indicator of mitochondrial function. Additionally, hydrogen gas, a PGC-1α activator, activates CoQ10 to rejuvenate exhausted CD8^+^ T cells, thereby enhancing the clinical efficacy of nivolumab [94]. Moreover, AMPK has been proven to drive mitochondrial biogenesis by promoting the activation of PGC-1α [95]. Metformin, an activator of AMPK, increases PGC-1α expression and restores mitochondrial FAO to promote mitochondrial function, thus reinvigorating CD8^+^ T cell exhaustion [96,97,98]. IL-15 can also induce PGC-1α expression and promote mitochondrial biogenesis [99].

### 4.2. Alleviating ROS Production

Neutralizing intracellular ROS can restore T cell self-renewal and reverse metabolic defects, as ROS accumulation limits cell proliferation and self-renewal gene expression in chronically infected T cells. Utilizing antioxidants (ROS scavengers) to restore T cell function offers a promising approach for rescuing exhausted T cell immunity. Antioxidants mitigate ROS production and reduce the effects of oxidative damage [100]. N-acetylcysteine (N-Ac) is a cell-permeable antioxidant that enhances glutathione synthesis to rescue the proliferation and effector function of exhausted T cells, thereby neutralizing intracellular ROS [75]. Treatment with N-Ac alone or in conjunction with anti-PD-L1 therapy enhances anti-tumor immunity [101]. Mitochondria-targeted antioxidants, such as mitoquinone (MitoQ), the piperidine nitroxide MitoTempo, or Trolox (a water-soluble Vitamin E analog), can reduce ROS production and effectively restore mitochondrial function in exhausted CD8 T cells [75,80]. Treatment with MitoTempo combined with ‘mitochondrial fission inhibitor’ mdivi-1, ‘fusion promoter’ M1, and IL-15 increases T cell metabolic fitness and restores the function of exhausted HIV-specific CD8 T cells [68].

Nicotinamide adenine dinucleotide (NAD) participates in redox reactions by transferring hydrogen ions and electrons, regulating ROS production, and maintaining cellular metabolic balance. Five precursor substances can be converted into NAD+, such as nicotinic acid (NA), nicotinamide (NAM), and nicotinamide riboside (NR) [102,103]. NR intervention mitigates CD4^+^ T cell exhaustion by upregulating silent information regulator 1 (SirT1) expression levels, which increases mitochondrial function [104]. Furthermore, the addition of NR to stimulate mitophagy can significantly alleviate mitochondrial ROS (mtROS) levels and relieve mitochondrial depolarization, thereby preventing exhaustion [57]. Similarly, supplementation with NAM reverses the expression of inhibitory receptors TIM-3 and LAG-3, restores the production of IL-2, IFN-γ, and TNF-α, and induces T cell differentiation towards effector memory and terminal effector states, ultimately inhibiting T cell exhaustion, which is associated with reducing ROS generation [105]. Furthermore, some studies have shown that metformin inhibits the production of intracellular ROS by promoting mitochondrial biogenesis through activation of the AMPK-PGC-1α pathway [106]. The immunotherapy system based on biomaterials has also enhanced the therapeutic effect [107]. Drug delivery nanoparticles exhibit multiple capabilities in the treatment and diagnosis of cancer [108]. For example, nanozymes with various activities can modulate immune responses by increasing or decreasing ROS levels around the chronic microenvironment [109,110].

### 4.3. Mitigating Hypoxia

Hypoxia is one of the main factors driving exhaustion, and hypoxic-mitigating therapy may be a feasible strategy for slowing terminal differentiation, which is beneficial for immunotherapy. Hypoxia is a common characteristic of the tumor microenvironment (TME), caused by abnormalities in vascular structure and function. Factors related to hypoxia mainly include high levels of vascular endothelial growth factor (VEGF) and prostaglandin E_2_ (PGE_2_) [111]. Axitinib, a potent tyrosine kinase inhibitor, exhibits nanomolar affinity for VEGF receptors (VEGFR) 1, 2, and 3. Using axitinib to mitigate hypoxia alleviates T-cell exhaustion and improves immunotherapy outcomes. Axitinib combined with PD-1 antibody has been approved by the Food and Drug Administration (FDA) for the treatment of advanced renal cell carcinoma [5]. Since hypoxia drives the CD39-dependent inhibitory function in exhausted T cells, limiting their effector functions, alleviating hypoxia by deleting NADH: ubiquinone oxidoreductase subunit S4, a mitochondrial complex I subunit, significantly reduces the suppression ability of exhausted cells [76].

### 4.4. Inducing ATP Production

Due to insufficient ATP during chronic infection, T-cell dysfunction arises. ATP can enhance mitochondrial function and mass, thereby reinvigorating these dysfunctional cells. As the CD39/CD73 pathway acts as a regulator of extracellular adenosine by scavenging ATP to adenosine, inhibiting CD39 and CD73 can enhance immune potency and reduce the percentage of exhausted T cells [112,113]. Blocking CD39 and CD73 in combination with immune checkpoint inhibitors and the chemotherapeutic drug oxaliplatin promotes anti-tumor immunity [114]. Furthermore, ubiquitin-specific proteases (USPs) also play a crucial role in regulating mitochondrial dynamics [115,116]. Ubiquitin carboxyl-terminal hydrolase 25 (USP25) interacts with ATP5a and ATP5b, promoting their stability and increasing ATP generation, thus maintaining mitochondrial morphology. USP25 deficiency, on the other hand, promotes T cell dysfunction through altered mitochondrial dynamics [117].

ATP synthase catalyzes ATP production and plays a significant role in the process of OXPHOS [118]. ATP synthase inhibitory factor 1 (ATPIF1) is an inhibitory protein of ATP synthase [119] and is crucial for maintaining mitochondrial structure [120]. ATPIF1 deficiency leads to tumor immune deficiency by inducing T-cell exhaustion, while ATPIF1 overexpression increases the density of mitochondrial cristae [121], thereby promoting mitochondrial OXPHOS and enhancing antitumor immunity [122].

### 4.5. Utilizing Mitochondrial Transfer

Mitochondrial transfer has emerged as a promising approach for the treatment of diseases, particularly metabolic disorders [61]. This process promotes the conversion of naïve T cells into effector and memory subsets, which are less prone to exhaustion during chronic *M. tuberculosis* infection. Given that T cells can acquire mitochondria from other cells via mitochondrial transfer, delivering normal mitochondria to exhausted T cells with damaged mitochondria may reverse impaired effector function and enhance immune efficacy [30]. Specifically, it regulates the expression of proteins associated with T cell exhaustion and drives the metabolic reprogramming of T cells from glycolysis to OXPHOS, significantly reducing T cell exhaustion in *M. tuberculosis*-infected mice [123].

Recent studies have emphasized the role of intercellular tunneling nanotube (TNTs)-mediated mitochondrial transfer in disease control. Baldwin and colleagues found that bone marrow stromal cells (BMSCs) construct nanotube connections with T cells and utilize these intercellular channels to transplant mitochondria from stromal cells into T cells [67]. TNTs-mediated mitochondrial transfer increases mitochondrial mass and fitness, and improves the antitumor efficacy of T cells, contributing to the prevention of T cell exhaustion [124]. Furthermore, the transferred mitochondria reduce the expression of mitochondrial fission protein Drp1 [125]. However, TNTs-mediated mitochondrial transfer has unidirectional and bidirectional patterns. During bidirectional mitochondrial transfer, the number of mitochondria transferred between the two types of cells varies, and the benefits obtained by the recipient cells also differ [126], indicating that TNTs-mediated mitochondrial transfer is regulated. Further studies are required to understand the regulatory mechanisms.

### 4.6. Other Strategies

Since the mitochondrial OXPHOS of exhausted T cells is impaired, measures to enhance OXPHOS can reverse T cell exhaustion. For instance, an extended half-life interleukin-10/Fc fusion protein (IL-10/Fc) has been shown to enhance mitochondrial OXPHOS. IL-10/Fc reprogrammed T cell metabolism by promoting OXPHOS in a mitochondrial pyruvate carrier (MPC)-dependent manner, reinvigorating terminally exhausted T cells and enhancing anti-tumor immunity [127,128].

Some studies discovered that specific long-chain fatty acids (LC-FAs) could damage the integrity and function of mitochondria in cytotoxic T lymphocytes (CTLs), leading to T cell exhaustion and impaired anti-tumor ability [129]. Not all LC-FAs are harmful to T cells. For instance, linoleic acid (LA) is the main positive regulator of CTL activity by improving metabolic fitness, inducing CTLs toward a memory phenotype, and preventing exhaustion [130]. Mechanistically, LA inserts into the membranes of mitochondria and endoplasmic reticulum (ER), leading to membrane remodeling and promoting membrane fusion between the two, forming a structure called mitochondria-ER contacts (MERCs) that regulate mitochondria function, control calcium (Ca^2+^) transport from the ER to mitochondria, and are critical for the activation, migration, and TCR signaling of T cells [131,132,133].

Furthermore, 5-aminolevulinic acid (5-ALA), a natural amino acid produced exclusively in mitochondria, has been shown to influence metabolic functions and works in conjunction with sodium ferrous citrate (SFC) to activate mitochondrial functions. The combination of 5-ALA/SFC may act synergistically with anti-PD-1/PD-L1 therapy to improve the anti-tumor efficacy of T cells [134]. Some studies have reported that inhibiting autophagy may be a potential mechanism for the efficacy of PD-1 blockade therapy in preventing T cell exhaustion [135].
ijms-26-07400-t001_Table 1Table 1Promising therapeutic strategies for regulating mitochondrial metabolism in reversing T-cell exhaustion.StrategyIntervention MethodsIntervention MechanismRef.Inducing the induction of PGC-1αBezafibrateA PGC-1α agonist that enhances mitochondrial biogenesis[88]C1QBP overexpressionAltering the impaired mitochondrial fitness[89]4-1BB agonistsPromoting PGC-1α-dependent mitochondrial fusion and biogenesis through activating p38-MAPK[92]Hydrogen gasA PGC-1α activator and activating CoQ10[94]MetforminActivating AMPK and directly inducing PGC-1α expression and restoring mitochondrial FAO[96,97,98]IL-15Inducing PGC-1α expression and promoting mitochondrial biogenesis[99]Alleviating ROS productionN-acetylcysteine (NAC)Neutralizing intracellular ROS[75,101]Mitoquinone (MitoQ)Reduce ROS production and effectively restore mitochondrial function[80]MitoTempo/TroloxAttenuate ROS production and effectively restore mitochondrial function[75,80]MitoTempo combined with mdivi-1, M1, and IL-15Increasing T cell metabolic fitness and restoring cell function[68]Nicotinamide nucleoside (NR)Alleviating mtROS levels and relieving depolarized mitochondria[57,104] Nicotinamide (NAM)Reducing ROS generation and increasing differentiation of effector T cells[105]MetforminInhibiting intracellular ROS production by promoting mitochondrial biogenesis[106]Mitigating hypoxiaAxitinibMitigating hypoxia[5]Deleting NADHAlleviating hypoxia[76]Inducing ATP productionCD39 and CD73 blockadePreventing the conversion of ATP to adenosine[114]USP25Increasing ATP generation and maintaining mitochondrial morphology[117]ATPIF1 overexpressionIncreasing the mitochondrial crista density and enhancing the mitochondrial OXPHOS[122]Utilizing mitochondrial transferTransporting normal mitochondria to exhausted T cellsReversing impaired effector function and enhancing immune efficacy[30,123]Intercellular TNTs-mediated mitochondrial transferIncreasing mitochondrial mass and fitness, and improving the antitumor efficacy of T cells[124]Other strategiesIL-10/Fc fusion proteinReprogramming T cell metabolism by promoting OXPHOS[127,128]Linoleic acid (LA)Enhancing the formation of MERCs, promoting memory differentiation, and preventing exhaustion[130]5-ALA/SFCActivating mitochondrial functions[134]Autophagy inhibitionInhibiting autophagy[135]

## 5. Conclusions and Perspectives

Mitochondria play a crucial role in cell proliferation, function, and differentiation. Mitochondrial metabolism is closely linked to mitochondrial morphology. The events of mitochondrial quality control mainly include mitochondrial biogenesis, fusion and fission, mitophagy, and mitochondrial transfer, enabling mitochondria to effectively respond to external stimuli and maintain mitochondrial dynamic stability. Exhausted T cells are often accompanied by mitochondrial loss and dysfunction. Targeting mitochondrial metabolism may be an important approach for rescuing T cell exhaustion and improving the efficacy of immunotherapy. However, a major limitation of mitochondrial transfer at present is its low efficiency (approximately 10%, with a maximum of 28%) and the lack of simple methods to track the transferred mitochondria, which currently hinders the translation of these results into clinical practice [67]. Mitochondrial transfer is also being studied in the treatment research of autoimmune diseases [136]. Furthermore, although the beneficial effects of CoQ10 or PGC-1α activators have been reported for the treatment of chronic diseases, differences in their effects have been observed in various studies, and further research is still needed to determine the optimal therapeutic dose.

Understanding the molecular mechanisms by which mitochondrial dysfunction promotes T cell exhaustion is crucial for developing new immunotherapy strategies. Further research into the regulation of mitochondrial metabolism during T cell exhaustion could provide novel approaches to combat T cell exhaustion. Exhaustion is not always detrimental. For instance, given the reduction in T cell exhaustion in autoimmunity, inducing exhaustion may serve as a therapeutic strategy for autoimmune diseases [137]. In rheumatoid arthritis (RA), T cells are unable to repair mitochondrial DNA [138]. The deficiency of DNA repair nuclease MRE11A in RA T cells impedes mitochondrial oxygen consumption and inhibits ATP production, whereas overexpression of MRE11A acts as a mitochondrial protector, restores mitochondrial fitness, and prevents tissue inflammation [139].

The intervention strategies for reversing T cell exhaustion through mitochondrial metabolism, such as metformin and bezafibrate, are discussed here. The current challenge of mitochondria-specific therapy lies in delivering bioactive molecules to mitochondria in vivo [140]. Furthermore, some mitochondria-targeting agents are known to have intrinsic toxicity, which limits their clinical applications [141]. Their application in T-cell exhaustion is still in the preclinical stage and requires further exploration of the optimal treatment dose and timing.

Metabolic reprogramming is a complex biological process involving multiple metabolic pathways and regulatory mechanisms. Biomarkers may help us better understand these mechanisms and how they are influenced by factors such as genetics, environment and lifestyle. By identifying key biomarkers, we can more precisely regulate metabolic processes, thereby achieving the goal of treating T-cell exhaustion. Therefore, biomarkers are needed to predict the response to metabolic reprogramming. In addition, extending the discovery from murine to human T cells requires overcoming multiple risks and challenges, including addressing species differences, ethical and regulatory issues, safety concerns, efficacy uncertainty, technical challenges, and feasibility issues in clinical practice. There is still a long way to go.

## Figures and Tables

**Figure 1 ijms-26-07400-f001:**
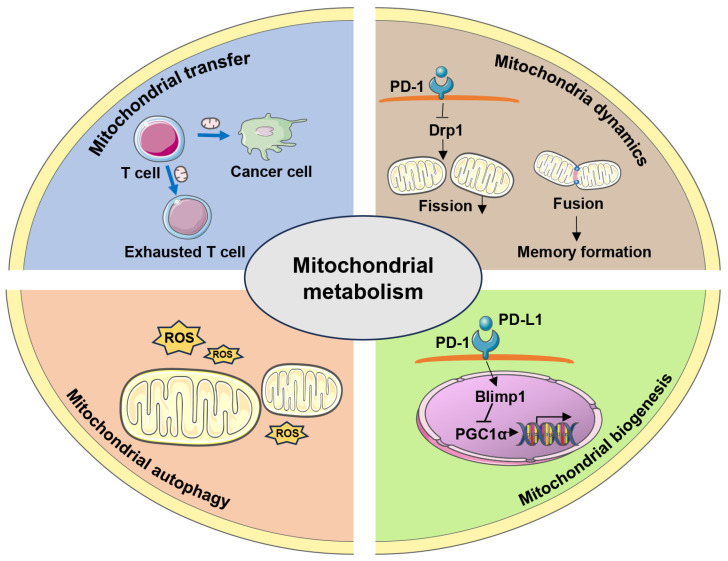
Effect of T cell mitochondrial quality control on immunometabolism. The differentiation and function of immune cells largely depend on specific metabolic programs determined by mitochondria. Mitochondrial metabolism is closely linked to mitochondrial morphology, which is influenced by mitochondrial biogenesis, dynamics, mitophagy, and mitochondrial transfer. Furthermore, mitochondrial biogenesis favors the metabolic reprogramming in T cells, with the crucial regulatory molecule being PGC-1α. PD-1 signaling promotes these metabolic alterations by driving Blimp1-mediated inhibition of PGC-1α. Mitochondrial dynamics, which include fusion and fission, determine T-cell differentiation and fate. Effector T cells possess fissed mitochondria with loosened cristae, whereas memory T cells have fused mitochondria with tighter cristae and expanded space. PD-1 signaling reduces the activity of mitochondrial fission protein Drp1; forcing mitochondrial fusion drives memory T cell differentiation. Mitochondrial autophagy (mitophagy) maintains T cell homeostasis. Mitochondrial transfer can alter the metabolic status of both donor and recipient by affecting mitochondrial mass, thus becoming a target for immunotherapy.

## Data Availability

Data sharing is not applicable.

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
