# Peer review of "Mitochondrial Metabolism in T-Cell Exhaustion"

_ijms, 2025, doi:10.3390/ijms26157400_

Round 1

Reviewer 1 Report

Comments and Suggestions for Authors

Summary of the Study

In this comprehensive review, the authors present an up-to-date report of the role of mitochondrial metabolism in T-cell exhaustion, specifically, in chronic infections and cancer. The authors outline key mitochondrial processes: biogenesis, dynamics (fusion/fission), mitophagy, and mitochondrial transfer, and how these regulate the functional fate of T cells. The manuscript is focused on exhausted CD8+ T cell subsets.

The authors also evaluate various therapeutic strategies aimed at restoring mitochondrial fitness to rejuvenate exhausted T cells, including PGC-1α activation, ROS modulation, hypoxia mitigation, and mitochondrial transfer, supported by an extensive reference list of recent literature.

Summary Critique

This review is well-written, thoroughly referenced, and highly relevant. It provides a valuable overview of mitochondrial quality control mechanisms in T cells and how their dysregulation contributes to exhaustion. It also integrates preclinical findings into potential translational interventions.

However, the manuscript could be strengthened by:

  1. Clarifying the categorization and signaling distinctions between progenitor vs. terminally exhausted T cells.
  2. Summarizing key mechanisms in concise schematic diagrams. The manuscript currently has only one figure. Here, it would be helpful for the reader to have available schematic diagrams of fundamental mechanisms.
  3. Streamlining redundancies in therapeutic strategy discussions.
  4. Including a more critical appraisal of limitations in current therapeutic approaches (e.g., efficiency of mitochondrial transfer, clinical viability of CoQ10 or PGC-1α activators).
  5. Appreciation that T cell exhaustion is a critical mechanism not only in tumor immunology, but also in autoimmune disease.

Point-by-Point Critique

Title and Abstract

Introduction (Lines 23–39)

Effectively frames the problem of T-cell exhaustion. The authors should define “memory T cell potential” explicitly (functional recall response, longevity, etc.).

Section 2: Mitochondrial Regulation of T-cell Function (Lines 64–154)

2.1 Biogenesis

Here, the emphasis lies on PGC-1α-dependent regulation and its therapeutic implications. Reference [20] and [32] are cited multiple times merge or streamline to avoid redundancy.

2.2 Dynamics

Excellent integration of fission/fusion and metabolic programming. The authors may want to summarize the contradictory roles of Drp1 (pro-fission vs. memory formation) in a Table or figure.

2.3 Mitophagy

Thorough discussion with appropriate focus on PINK1/Parkin. Please explain the link between mitophagy suppression and epigenetic reprogramming (cited from [54]) with more mechanistic detail.

2.4 Mitochondrial Transfer

Timely and exciting topic. Current limitations (e.g., efficiency, tracking difficulty) are only briefly noted in the conclusion, please introduce these caveats earlier in this section.

Section 3: Metabolic Features of Exhausted T Cells (Lines 155–212)

Clearly outlines metabolic deficiencies in exhausted T cells. Consider reordering this section: start with PD-1-driven metabolic effects, then move into mitochondrial depolarization and dysfunction. Add a table summarizing metabolic markers (e.g., ROS↑, ATP↓, PGC-1α↓) in terminally exhausted vs. progenitor exhausted T cells.

The authors may want to consider connecting to work done in autoimmune disease, which shows the critical importance of metabolic exhaustion in pathogenetic T cells and the benefit of mitochondrial transfer (e.g. PMID: 31327667, PMID: 34811544, PMID: 33868292, PMID: 39800938)

Section 4: Therapeutic Interventions (Lines 213–354)

The subsections are quite long and at times repetitive. Suggest summarizing each therapeutic mechanism with a simplified table or visual summary for clarity. Several agents (e.g., metformin, bezafibrate) are well-known mitochondrial enhancers, but their clinical application in T-cell exhaustion is still preclinical. Please include a comment on the current clinical development stage of these interventions.

Table 1: Therapeutic Strategies

Useful consolidation of agents and mechanisms. Consider adding a column indicating preclinical vs. clinical validation with reference. Please ensure that all referenced figures are present and clearly labeled.

Conclusions (Lines 358–373)

Concludes effectively and reinforces the review's relevance. Please include a more critical perspective on gaps in current understanding, especially: 1. The complexity of mitochondrial-targeting therapies in vivo. 2. Risk of generalizing findings from murine to human T cells. 3. Need for biomarkers to predict responsiveness to metabolic reprogramming.

References

You may consider reducing redundancy by consolidating closely related citations (e.g., PD-1 effects on PGC-1α, cristae morphology, and ROS).

Author Response

Dear Editor, Thanks for your careful consideration of our manuscript. We appreciate the helpful comments from reviewers very much. We have made careful revisions accordingly. Our point-by-point responses are as follows. Reviewer 1 This review is well-written, thoroughly referenced, and highly relevant. It provides a valuable overview of mitochondrial quality control mechanisms in T cells and how their dysregulation contributes to exhaustion. It also integrates preclinical findings into potential translational interventions. However, the manuscript could be strengthened by: 1. Clarifying the categorization and signaling distinctions between progenitor vs. terminally exhausted T cells. Answer: Thanks for your great suggestion. ‘Progenitor exhausted’ cells retain polyfunctionality, persist long term, and differentiate into ‘terminally exhausted’ TILs, while progenitor exhausted CD8+ TILs are better able to control tumor growth than are terminally exhausted T cells. Progenitor exhausted TILs can respond to anti-PD-1 therapy, but terminally exhausted TILs cannot [1]. Notably, the transcription factor TCF1 plays a cell-intrinsic and essential role in the generation of this CD8+ T cell subset. Progenitor exhausted CD8+ T cells also express the chemokine receptor CXCR5. CXCR5− CD8+ T cells are terminally differentiated with limited proliferative potential, whereas the CXCR5+ CD8+ T cells act as stem cells, undergo a slow self-renewal, and also give rise to the more terminally differentiated effector-like CD8+ T cell subset. Furthermore, the CXCR5+Tim-3− subset is TCF1+, whereas the CXCR5−Tim-3+ cells are TCF1−. CXCR5+ CD8+ T cells expressed several genes in the Wnt signalling pathway that are known to be associated with self-renewal and the maintenance of haematopoietic stem cells [3]. We have added the above categorization and signaling distinctions between progenitor vs. terminally exhausted T cells in the manuscript to make it clearer. [1] Miller, B.C., Sen, D.R., Al Abosy, R. et al. Subsets of exhausted CD8+ T cells differentially mediate tumor control and respond to checkpoint blockade. Nat Immunol 20, 326–336 (2019). [2] Im SJ, Hashimoto M, Gerner MY, Lee J, Kissick HT, Burger MC, Shan Q, Hale JS, Lee J, Nasti TH, Sharpe AH, Freeman GJ, Germain RN, Nakaya HI, Xue HH, Ahmed R. Defining CD8+ T cells that provide the proliferative burst after PD-1 therapy. Nature. 2016 Sep 15;537(7620):417-421. [3] Reya T, et al. A role for Wnt signalling in self-renewal of haematopoietic stem cells. Nature. 2003;423:409–414. 2. Summarizing key mechanisms in concise schematic diagrams. The manuscript currently has only one figure. Here, it would be helpful for the reader to have available schematic diagrams of fundamental mechanisms. Answer: Thanks for your constructive comments. You're right, using concise schematic diagrams to summarize key mechanisms enhances understandability. However, the mechanisms are quite complex, and it is not convenient to summarize them into a single diagram. Therefore, an ideal diagram was not created. 3. Streamlining redundancies in therapeutic strategy discussions. Answer: Thanks for your comments. We have eliminated the redundant content in the discussion of treatment strategies. 4. Including a more critical appraisal of limitations in current therapeutic approaches (e.g., efficiency of mitochondrial transfer, clinical viability of CoQ10 or PGC-1α activators). Answer: Thanks for your suggestion. Yes, a major limitation of mitochondrial transfer at present is its low efficiency (approximately 10%, with a maximum of 28%) and the lack of simple methods to track the transferred mitochondria, which currently hinders the translation of these results into clinical practice. Although the beneficial effects of CoQ10 or PGC-1α activators have been reported for the treatment of chronic diseases, differences in their effects have been observed in various studies, indicating that further research is still needed to determine the optimal therapeutic dose. We have added these ideas in the ‘5. Conclusions and Perspectives’ section of the manuscript to enhance its quality. 5. Appreciation that T cell exhaustion is a critical mechanism not only in tumor immunology, but also in autoimmune disease. Answer: Thanks for your constructive comments. T-cell exhaustion also plays a central role in determining outcome in autoimmune disease. Given the reduction of T cell exhaustion in autoimmunity, inducing exhaustion may be a therapeutic strategy for autoimmune diseases [1]. We have added these ideas to the manuscript to make it better. [1] McKinney EF, Lee JC, Jayne DR, Lyons PA, Smith KG. T-cell exhaustion, co-stimulation, and clinical outcome in autoimmunity and infection. Nature. 2015 Jul 30;523(7562):612-6. 6. Some related works are encouraged to be cited: Exploration, 2022, 2:20210157; Chin. Chem. Lett. 2023, 34, 107518. Answer: Thanks for your suggestion. We carefully read the above literature and cite these studies in the manuscript to make this review clearer. Point-by-Point Critique Title and Abstract Introduction (Lines 23–39) Effectively frames the problem of T-cell exhaustion. The authors should define “memory T cell potential” explicitly (functional recall response, longevity, etc.). Answer: We explained the meaning of “memory T cell potential” in the text. Memory T cell potential includes the antigen-independent self-renewal ability and the capacity to generate strong recall responses. Section 2: Mitochondrial Regulation of T-cell Function (Lines 64–154) 2.1 Biogenesis Here, the emphasis lies on PGC-1α-dependent regulation and its therapeutic implications. Reference [20] and [32] are cited multiple times merge or streamline to avoid redundancy. Answer: We have removed references [20] and [32] in certain places in the manuscript to avoid redundancy. 2.2 Dynamics Excellent integration of fission/fusion and metabolic programming. The authors may want to summarize the contradictory roles of Drp1 (pro-fission vs. memory formation) in a Table or figure. Answer: Thanks. Mitochondrial fission is governed by the dynamin-related protein 1 (Drp1), whereas optic atrophy 1 (OPA1), mitofusin 1 (MFN1), and mitofusin 2 (MFN2) favor mitochondrial fusion and promote cellular ATP generation. Therefore, treating T cells with the mitochondrial fission inhibitor mdivi-1 and fusion promoter M1 induces mitochondrial fusion, thereby conferring a memory T cell phenotype and promoting the generation of memory-like T cells. We have made some changes in Figure 1 to enhance clarity. 2.3 Mitophagy Thorough discussion with appropriate focus on PINK1/Parkin. Please explain the link between mitophagy suppression and epigenetic reprogramming (cited from [54]) with more mechanistic detail. Answer: Thanks. We carefully read the original literature and found that the statements in the text were inaccurate. Mitophagy suppression is not directly relevant to epigenetic reprogramming. The article suggests that terminally exhausted CD8+ T cells have been shown to exhibit distinct epigenetic landscapes, including chromatin accessibility and DNA methylation patterns. Defective mitophagy or impaired oxidative phosphorylation triggers mitochondrial reactive oxygen species production, which in turn promotes a T cell exhaustion program. Forcing depolarized mitochondria accumulation by disturbing mitochondrial dynamics reinforces phenotypic and epigenetic reprogramming for T cell exhaustion [1]. We have revised the relevant statements in the text to make them more accurate. [1] Li W, Cheng H, Li G, Zhang L. Mitochondrial Damage and the Road to Exhaustion. Cell Metab. 2020 Dec 1;32(6):905-907. 2.4 Mitochondrial Transfer Timely and exciting topic. Current limitations (e.g., efficiency, tracking difficulty) are only briefly noted in the conclusion, please introduce these caveats earlier in this section. Answer: Thanks. We have added these limitations in the earlier section of the manuscript. Section 3: Metabolic Features of Exhausted T Cells (Lines 155–212) Clearly outlines metabolic deficiencies in exhausted T cells. Consider reordering this section: start with PD-1-driven metabolic effects, then move into mitochondrial depolarization and dysfunction. Add a table summarizing metabolic markers (e.g., ROS↑, ATP↓, PGC-1α↓) in terminally exhausted vs. progenitor exhausted T cells. The authors may want to consider connecting to work done in autoimmune disease, which shows the critical importance of metabolic exhaustion in pathogenetic T cells and the benefit of mitochondrial transfer (e.g. PMID: 31327667, PMID: 34811544, PMID: 33868292, PMID: 39800938) Answer: Thanks for your comments. (1) We have reordered this section according to your suggestion to enhance its logical clarity. (2) Metabolically, progenitor exhausted T cells use mitochondrial FAO and OXPHOS for energy, whereas terminally exhausted T cells primarily rely on glycolytic metabolism, with impaired glycolysis and OXPHOS. The exhausted T cells mentioned in this review mainly refer to terminally exhausted T cells. Therefore, we did not add a table summarizing metabolic markers (e.g., ROS↑, ATP↓, PGC-1α↓) in terminally exhausted vs. progenitor exhausted T cells. (3) We carefully read the above literature and cite these studies in the manuscript to make this review clearer. Section 4: Therapeutic Interventions (Lines 213–354) The subsections are quite long and at times repetitive. Suggest summarizing each therapeutic mechanism with a simplified table or visual summary for clarity. Several agents (e.g., metformin, bezafibrate) are well-known mitochondrial enhancers, but their clinical application in T-cell exhaustion is still preclinical. Please include a comment on the current clinical development stage of these interventions. Answer: Thanks. We have added a comment on the current clinical development stage of these interventions to make this review more comprehensive. Table 1: Therapeutic Strategies Useful consolidation of agents and mechanisms. Consider adding a column indicating preclinical vs. clinical validation with reference. Please ensure that all referenced figures are present and clearly labeled. Answer: Thanks for your constructive suggestion. You're right. Although several clinical studies on mitochondrial-targeting antioxidants have been performed, especially those on MitoQ. However, the intervention agents (e.g., metformin, bezafibrate) in intervening T-cell exhaustion have been studied in vitro and in vivo, yet clinical studies are lacking. All are in the exploration stage, so we added some words to the discussion section. Conclusions (Lines 358–373) Concludes effectively and reinforces the review's relevance. Please include a more critical perspective on gaps in current understanding, especially: 1. The complexity of mitochondrial-targeting therapies in vivo. 2. Risk of generalizing findings from murine to human T cells. 3. Need for biomarkers to predict responsiveness to metabolic reprogramming. Answer: Thanks for your suggestion. We have carefully read the above information, the relevant literature, and added these ideas to the manuscript to make this review clearer. References You may consider reducing redundancy by consolidating closely related citations (e.g., PD-1 effects on PGC-1α, cristae morphology, and ROS). Answer: Thanks for your suggestion. We have merged closely related references to reduce redundancy. We have revised our manuscript very carefully. If there are any questions, please let us know and we will do more revisions. We are looking forward to hearing from you. Best wishes, Fei Li, M. D.

Reviewer 2 Report

Comments and Suggestions for Authors

Reviewer’s Comments

The manuscript by Fei Li et al “Mitochondrial metabolism in T-cell exhaustion” (3768174) is a high-quality and timely review article that advances understanding of mitochondrial metabolism in T-cell exhaustion. With minor revisions to clarify contradictions, update references, and enhance clinical relevance, this review article could be published in the International Journal of Molecular Sciences.

  1. Main Question Addressed by the Article

The article comprehensively addresses the role of mitochondrial metabolism in T-cell exhaustion, focusing on mechanisms (biogenesis, fission/fusion, mitophagy, transfer) and therapeutic strategies to reverse exhaustion. The central question—how mitochondrial dysfunction drives T-cell exhaustion and how targeting it can enhance immunotherapy—is clearly articulated.

  1. Originality/Novelty
  • Strengths: The review synthesizes cutting-edge research on mitochondrial dynamics in T-cell exhaustion, a rapidly evolving field. It highlights understudied areas like mitochondrial transfer and PGC-1α’s role.
  • Gaps: While thorough, it could better differentiate its contribution from prior reviews (e.g., by emphasizing recent breakthroughs like nanotube-mediated mitochondrial transfer).
  1. Significance of Content
  • The content is highly significant, covering metabolic reprogramming, mitochondrial dysfunction markers (e.g., ROS, ATP depletion), and therapeutic interventions (e.g., PGC-1α agonists, antioxidants).
  • Suggestion: Add a table summarizing key mitochondrial alterations in exhausted vs. functional T cells for clarity.
  1. Scientific Soundness
  • The review is well-supported by recent studies (e.g., Scharping 2021, Yu 2020) and balances mechanistic insights with clinical implications.
  • Minor concern: Some claims (e.g., Drp1’s dual role in exhaustion/memory) need clearer reconciliation.
  1. Contribution Compared to Existing Literature
  • This review stands out by integrating mitochondrial dynamics (fission/fusion, transfer) with metabolic reprogramming and immunotherapy. It updates older reviews with 2023–2024 references (e.g., IL-10/Fc, linoleic acid).
  • Improvement: Contrast findings with seminal papers (e.g., Wherry 2015) to highlight advancements.
  1. Consistency of Conclusions
  • Conclusions align with evidence, emphasizing mitochondrial restoration as a therapeutic axis. However, the clinical translatability of mitochondrial transfer (noted as low-efficiency) could be discussed more critically.
  1. Relevance of Background References
  • References are up-to-date and authoritative (e.g., Pearce 2013, Buck 2016). A few older citations (e.g., Vega 2000) could be replaced with recent reviews on PGC-1α.
  1. Overall Design of the Review
  • Logical flow: Introduction → Mechanisms → Therapeutic strategies → Conclusions.
  • Suggestion: Improve subheading hierarchy (e.g., merge Sections 2.1–2.4 under "Mitochondrial Quality Control").
  1. Visuals

This review article includes only one figure, and its illustrations do not accurately represent the processes described in the figure legend. This discrepancy is particularly evident in the depictions of mitochondrial dynamics, mitochondrial autophagy, and mitochondrial transfer. The inner mitochondrial membrane (IMM) undergoes dramatic changes in overall shape, density, and cristae morphology prior to autophagy and during fission and fusion—yet these dynamic transformations are not illustrated in the figure. In fact, the current depiction only represents the IMM morphology of healthy mitochondria. Furthermore, as healthy T cells transition into either cancerous or exhausted states, their cristae and IMM structures undergo significant remodeling. Given the functional importance of these morphological changes, accurate visual representations should be included to reflect both (1) the dynamic processes of mitochondrial fission/fusion/autophagy and (2) the structural shifts linked to T cell dysfunction.

  1. Interest to Readers
  • The article will appeal to immunologists, oncologists, and translational researchers. Clinical applications (e.g., Table 1) enhance relevance.
  • Enhancement: Add a "Future Directions" subsection to discuss unresolved questions (e.g., biomarkers for mitochondrial fitness).
  1. Quality of English Language
  • Generally excellent, but minor grammatical errors exist (e.g., "mitochondrial biosynthesis" → "biogenesis" for consistency).

Recommendation: Minor Revision (to address clarifications and structural tweaks)

The review article by Fei Li et al “Mitochondrial metabolism in T-cell exhaustion” (3768174) advances understanding of mitochondrial metabolism in T-cell exhaustion and is a valuable contribution to the field. I recommend this article for publication in the International Journal of Molecular Sciences provided the specific points for revision are appropriately addressed.

Specific Points for Revision

  1. Clarify contradictions: Reconcile Drp1’s role in exhaustion (Section 3) vs. memory (Section 2.2).
  2. Update references: Replace outdated citations (e.g., Vega 2000) with recent reviews on PGC-1α.
  3. Improve visuals: Expand Figure 1 to include exhaustion-specific pathways (e.g., PD-1’s impact on cristae and overal inner mitochondrial membrane morphology – see section 9 above for details).
  4. Enhance clinical context: Discuss challenges in translating mitochondrial transfer (Section 5) and ongoing trials.
  5. Proofread: Fix minor typos (e.g., "TGC-1α" → "PGC-1α" in Figure 1 caption).

Author Response

Dear Editor,

Thanks for your careful consideration of our manuscript. We appreciate the helpful comments from reviewers very much. We have made careful revisions accordingly. Our point-by-point responses are as follows.

Reviewer 2

The manuscript by Fei Li et al “Mitochondrial metabolism in T-cell exhaustion” (3768174) is a high-quality and timely review article that advances understanding of mitochondrial metabolism in T-cell exhaustion. With minor revisions to clarify contradictions, update references, and enhance clinical relevance, this review article could be published in the International Journal of Molecular Sciences.

  1. Main Question Addressed by the Article

The article comprehensively addresses the role of mitochondrial metabolism in T-cell exhaustion, focusing on mechanisms (biogenesis, fission/fusion, mitophagy, transfer) and therapeutic strategies to reverse exhaustion. The central question—how mitochondrial dysfunction drives T-cell exhaustion and how targeting it can enhance immunotherapy—is clearly articulated.

  1. Originality/Novelty
  • Strengths: The review synthesizes cutting-edge research on mitochondrial dynamics in T-cell exhaustion, a rapidly evolving field. It highlights understudied areas like mitochondrial transfer and PGC-1α’s role.
  • Gaps: While thorough, it could better differentiate its contribution from prior reviews (e.g., by emphasizing recent breakthroughs like nanotube-mediated mitochondrial transfer).

Answer: Thanks for your constructive comments. We added some words to emphasize the importance of nanotube-mediated mitochondrial transfer in disease control.

  1. Significance of Content
  • The content is highly significant, covering metabolic reprogramming, mitochondrial dysfunction markers (e.g., ROS, ATP depletion), and therapeutic interventions (e.g., PGC-1α agonists, antioxidants).
  • Suggestion: Add a table summarizing key mitochondrial alterations in exhausted vs. functional T cells for clarity.

Answer: Thanks. We have added more words on the key mitochondrial alterations in exhausted vs. functional T cells to make it clearer.

  1. Scientific Soundness
  • The review is well-supported by recent studies (e.g., Scharping 2021, Yu 2020) and balances mechanistic insights with clinical implications.
  • Minor concern: Some claims (e.g., Drp1’s dual role in exhaustion/memory) need clearer reconciliation.

Answer: Thanks for your suggestion. We have carefully read the relevant literature about Drp1 and made some changes in the manuscript to make this review clearer.

  1. Contribution Compared to Existing Literature
  • This review stands out by integrating mitochondrial dynamics (fission/fusion, transfer) with metabolic reprogramming and immunotherapy. It updates older reviews with 2023–2024 references (e.g., IL-10/Fc, linoleic acid).
  • Improvement: Contrast findings with seminal papers (e.g., Wherry 2015) to highlight advancements.

Answer: Thanks for your suggestion. Here, we mainly want to emphasize the changes in mitochondrial metabolism in T-cell exhaustion. This manuscript also cites Wherry's 2015 research in the exhaustion definition section at the beginning.

  1. Consistency of Conclusions
  • Conclusions align with evidence, emphasizing mitochondrial restoration as a therapeutic axis. However, the clinical translatability of mitochondrial transfer (noted as low-efficiency) could be discussed more critically.

Answer: Thanks for your suggestion. We have added some words in the main text to discuss the current problems of mitochondrial transfer.

  1. Relevance of Background References
  • References are up-to-date and authoritative (e.g., Pearce 2013, Buck 2016). A few older citations (e.g., Vega 2000) could be replaced with recent reviews on PGC-1α.

Answer: Thanks for your suggestion. We replaced the older citation with recent reviews on PGC-1α.

  1. Overall Design of the Review
  • Logical flow: Introduction → Mechanisms → Therapeutic strategies → Conclusions.
  • Suggestion: Improve subheading hierarchy (e.g., merge Sections 2.1–2.4 under "Mitochondrial Quality Control").

Answer: Thanks for your suggestion. Mitochondrial quality control mainly includes mitochondrial biogenesis, fusion and fission, mitophagy, and mitochondrial transfer. The merged subheading may not be as clear as the current separate subheading, so the previous wording has been retained.

  1. Visuals

This review article includes only one figure, and its illustrations do not accurately represent the processes described in the figure legend. This discrepancy is particularly evident in the depictions of mitochondrial dynamics, mitochondrial autophagy, and mitochondrial transfer. The inner mitochondrial membrane (IMM) undergoes dramatic changes in overall shape, density, and cristae morphology prior to autophagy and during fission and fusion—yet these dynamic transformations are not illustrated in the figure. In fact, the current depiction only represents the IMM morphology of healthy mitochondria. Furthermore, as healthy T cells transition into either cancerous or exhausted states, their cristae and IMM structures undergo significant remodeling. Given the functional importance of these morphological changes, accurate visual representations should be included to reflect both (1) the dynamic processes of mitochondrial fission/fusion/autophagy and (2) the structural shifts linked to T cell dysfunction.

  1. Interest to Readers
  • The article will appeal to immunologists, oncologists, and translational researchers. Clinical applications (e.g., Table 1) enhance relevance.
  • Enhancement: Add a "Future Directions" subsection to discuss unresolved questions (e.g., biomarkers for mitochondrial fitness).
  1. Quality of English Language
  • Generally excellent, but minor grammatical errors exist (e.g., "mitochondrial biosynthesis" → "biogenesis" for consistency).

Recommendation: Minor Revision (to address clarifications and structural tweaks)

The review article by Fei Li et al “Mitochondrial metabolism in T-cell exhaustion” (3768174) advances understanding of mitochondrial metabolism in T-cell exhaustion and is a valuable contribution to the field. I recommend this article for publication in the International Journal of Molecular Sciences provided the specific points for revision are appropriately addressed.

Specific Points for Revision

  1. Clarify contradictions: Reconcile Drp1’s role in exhaustion (Section 3) vs. memory (Section 2.2).

Answer: Thanks for your suggestion. We have carefully read the relevant literature about Drp1, and made some changes to the manuscript to make this review clearer.

  1. Update references: Replace outdated citations (e.g., Vega 2000) with recent reviews on PGC-1α.

Answer: Thanks for your suggestion. We replaced the older citation with recent reviews on PGC-1α.

  1. Improve visuals: Expand Figure 1 to include exhaustion-specific pathways (e.g., PD-1’s impact on cristae and overal inner mitochondrial membrane morphology – see section 9 above for details).

Answer: Thanks. We have added some details about the changes in mitochondrial morphology in the figure.

  1. Enhance clinical context: Discuss challenges in translating mitochondrial transfer (Section 5) and ongoing trials.

Answer: Thanks. We have added more details about discuss challenges in mitochondrial transfer.

  1. Proofread: Fix minor typos (e.g., "TGC-1α" → "PGC-1α" in Figure 1 caption).

Answer: Thanks for your suggestion. We have corrected the word in the manuscript to make it clearer.

We have revised our manuscript very carefully. If there are any questions, please let us know and we will do more revisions. We are looking forward to hearing from you.

Best wishes,

Fei Li, M. D.